# An SEM-REM-Based Study on the Driving and Restraining Mechanisms and Potential of Reclaimed Water Utilization in China

**Jintao Zheng [1,2,3], Jiufu Liu [1,2], Tao Ma [1,2,*], Anbang Peng [1,2] and Xiyuan Deng [1,2]**

1   Nanjing Hydraulic Research Institute, Nanjing 210029, China; weigan0831@163.com (J.Z.); tuxian43105@163.com (J.L.); kongshan1552981@163.com (A.P.); jia69777522@163.com (X.D.)
2   State Key Laboratory of Hydrology-Water Resources and Hydraulic Engineering, Nanjing 210029, China
3   College of Hydrology and Water Resources, Hohai University, Nanjing 210098, China
*   Correspondence: taomamama@163.com

**Abstract:** In order to promote the efficient use of reclaimed water in China and make water resources allocation better structured, this paper analyzed the factors that drive and restrain the current utilization of reclaimed water and unveiled their correlation and hierarchy in a way to develop a non-recursive structural framework of what drives and restrain reclaimed water use. By structural equation modeling (SEM), the transmission path of affecting factors was identified, the contribution of the factors quantified, and key indicators for potential prediction selected. On that basis, a random-effects model (REM) was built to predict the potential availability of the country's reclaimed water. Meanwhile, parametric confidence intervals at 10–90% quantile levels were described, given the uncertainty of REM parameters. The results showed that four indicators for potential prediction, namely the total amount of wastewater treated, the density of water pipelines in built-up areas, investment in facilities for reclaimed water treatment, and the processing of applications for water treatment patents, are intertwined with the utilization of reclaimed water. Overall, the REM for potential prediction produced more precise fitting results, with the most significant fitting error standing at 5.9%. Going ahead, China is set to maintain the rapid growth in reclaimed water use, and up to 13.7 billion cubic meters of reclaimed water is expected to be available by 2025. This will help better structure the urban water supply and render regional water recycling more efficient.

**Keywords:** reclaimed water; potential analysis; planning; water resources management

## 1. Introduction

Despite its abundant water resources, China has a big population, with average per capita water resources amounting to only one-fourth of the world's average [1]. China, thus, represents one of the countries grappling with water scarcity. Moreover, with accelerated industrialization and explosive urbanization in the country, water pollution has become increasingly severe, leading to intensified pressure of the water shortage [2]. Finding how to better respond to water shortages is imperative for China at this stage, as the gap between water supply and demand grows. The dilemma can be addressed by utilizing reclaimed water, which can substitute for conventional water resources, reduce people's dependence on freshwater, and increase the efficiency of water recycling. "Including reclaimed water into the unified system for water resources allocation" has become a critical initiative of China's Ministry of Water Resources in a new period to promote the water-saving program nationwide and implement the principle of prioritizing water conservation [3]. However, the reclaimed water use in China remains at the development stage [4], with the proportion of reclaimed water in the country's total water supply at less than 1.5% [5], far lower than the advanced world levels [6]. Therefore, efforts should be made to research the factors that drive and restrain the utilization of reclaimed water, analyze the relationship between

the utilization scale and these factors and the relevant response mechanism, and identify the potential availability of reclaimed water. Doing so is imperative to scale up the use of reclaimed water and drive its efficiency, while eliminating the bottleneck that restricts reclaimed water utilization.

Previous research on the driving and restraining mechanisms for reclaimed water utilization concentrated on the macroscopic and quantitative analysis of the interplay between industries or actors [7–9]. Based on the principles of environmental economics, Wang et al. [10] made a qualitative analysis of the correlation between actors utilizing reclaimed water before constructing a preliminary framework of incentive mechanisms for reclaimed water utilization. Dong et al. [11] believed that examining the market price correlation in a science-based manner and innovating the price mechanism of reclaimed water resources are crucial for the quality development of reclaimed water resources, but their research on the pricing mechanism for reclaimed water did not go further. Using a system dynamics approach, Lin et al. [12] simulated the changes in water supply and demand under such regulation schemes as quota adjustment and unconventional water use and conducted a quantitative analysis of water-saving effects under different incentive mechanisms, which, however, was considered as a subjective approach.

At the same time, few studies were conducted on the potential availability of reclaimed water, while more efforts were made to analyze supply and demand balancing for a given time period, when such conditions as regional planning, economic and technical realities are certain [13–15]. Tang et al. [16] identified the potential availability of reclaimed water by examining the interplay of the available reclaimed water in Yinchuan, the capital city of North Central China's Ningxia Hui Autonomous Region, which was calculated according to the treatment capacity of reclaimed water plants, and the reclaimed water demand based on the fields in need of such kind of water. Building on regional planning for water supply and a quota-based method, Hochstrat et al. [17] calculated water demand in urban areas and measured the potential availability of reclaimed water by determining the proportion of reclaimed water utilization over some specified future time period. Supply and demand balancing can be analyzed with sufficient measured data on water supply and demand volumes in a studied area. However, calculating the potential availability of reclaimed water only based on supply–demand difference or some water demand ratios requires massive data of multifarious types. The selection of ratios of reclaimed water to the total water supply is subjective to a certain extent. Moreover, that approach makes it challenging to identify the impacts of factors such as techniques and market on reclaimed water utilization.

In summary, the problem facing current research efforts on the potential of reclaimed water development and utilization is two-fold. First, existing works on the driving and restraining mechanisms of reclaimed water utilization are largely macroscopic [7–9]. As a result, many influencing factors can only qualitatively judge whether it is beneficial or unfavorable to the development and utilization of reclaimed water. For example, the higher the regional economy and technology level, the more beneficial the development and utilization of reclaimed water [18], but how significant is this impact? This is a complex problem to solve. Second, conducting a supply–demand balancing analysis requires massive data of multifarious types. It is necessary to have the total water consumption and water demand data of various industries related to the field of reclaimed water utilization in order to predict the potential of reclaimed water development and utilization [19]. However, how can we calculate reclaimed water's development and utilization potential for areas lacking primary data? In response to the above problems, first of all, a structural equation model (SEM) was built in the present work to describe in detail the transmission pathways between the drivers and restraints of reclaimed water utilization and quantify the interaction intensity between the driving and limiting factors. Then, in the present work, the contribution rate of influencing factors to the development and utilization of reclaimed water was analyzed, from which the key indicators ($\leq$4) were selected. The selected indicators and the measured data of reclaimed water development and utilization

were combined to establish a random-effects model (REM) to perform a regression analysis, and forecast the development and utilization of reclaimed water.

This paper aims to explain the driving and constraint mechanisms of the development and utilization of reclaimed water, to clarify the response relationship between the scale of development and utilization and the driving factors and constraints, to screen out the key indicators that affect the development and utilization of reclaimed water, and to construct a prediction model for the development and utilization potential of reclaimed water suitable for areas with few data. It is expected that this paper could provide a new perspective for the efficient utilization of reclaimed water and offer a quantitative basis for the coordination between development and utilization of reclaimed water.

## 2. Research Method

### 2.1. Structural Equation Modeling

Structural equation modeling, or SEM [20], is a covariance-matrix-based statistical method to analyze the relationship between variables, in which abstract concepts can be estimated quantitatively and verified. Reclaimed water utilization is related to natural resources, the environment, and economic and social factors. Moreover, the interconnectivity between the utilization scale and driving and restraining factors, and their response mechanisms, involves multiple latent variables that cannot be observed directly, such as the technical strength in reclaimed water treatment and regional economic growth. Therefore, to clarify the correlation between latent variables and reclaimed water use, this paper selected SEM to analyze the driving and restraining mechanisms for utilizing reclaimed water.

Structural equation modeling (SEM) comprises two parts: a measurement model and a structural model. The former, consisting of latent and measured variables, describes the relationship between measured and latent variables, and quantifies the latent variables that cannot be directly observed. The general equation of the measurement model is [21]:

$$y = \Lambda_y \eta + \varepsilon \tag{1}$$

$$x = \Lambda_x \xi + \delta \tag{2}$$

where $y$ is the endogenous measured variable; $\eta$ is the endogenous latent variable; $\Lambda_y$ indicates the coefficient of correlation between endogenous measured and latent variables; $\varepsilon$ is the residual of the endogenous measured variable not fully explained; $x$ is the exogenous measured variable; $\xi$ is the exogenous latent variable; $\Lambda_x$ suggests the coefficient of correlation of exogenous measured and latent variables; $\delta$ is the residual of the exogenous measured variable not fully explained.

The structural model can describe the causal association between latent variables and explore potential stratification among variables. The general equation of the structural model is [19]:

$$\eta = B\eta + \Gamma\xi + \varsigma \tag{3}$$

where $B$ is the coefficient of relationship between endogenous latent variables; $\Gamma$ is the correlation coefficient of the impacts of exogenous latent variables on endogenous ones; $\xi$ denotes the residual term in the equation.

The basic procedure of the SEM analysis can be summarized into two stages: model development stage and model evaluation stage. The primary purpose of the model development stage (1st stage) is to establish a hypothetical model or framework, which is suitable for SEM analysis concepts and technical requirements. This stage involves the two concepts of theory development and model definition and recognition. Although these two concepts are represented in a continuous process in Figure 1, their relationship only illustrates the sequence of concept occurrence. In practice, the operation of these two concepts is an interactive and reciprocating process. Once the development of the SEM model is completed, the researcher needs to collect actual measurement data to test the appropriateness of the proposed conceptual model. This stage is called the model

estimation and evaluation stage (2nd stage). It begins with the establishment of the sample and the progress of the measurement and the obtained observation data are processed and estimated according to the requirements of the SEM analysis tool. It is worth noting that in the 2nd stage, SEM analysis tools usually provide measurement information for model adjustment and modification. Researchers can adjust the proposed hypothesis model based on these statistical test results and repeat the parameter estimation and model evaluation. This process will lead researchers to continue to derive more meaningful models or hypotheses and re-propose a more reasonable SEM model.

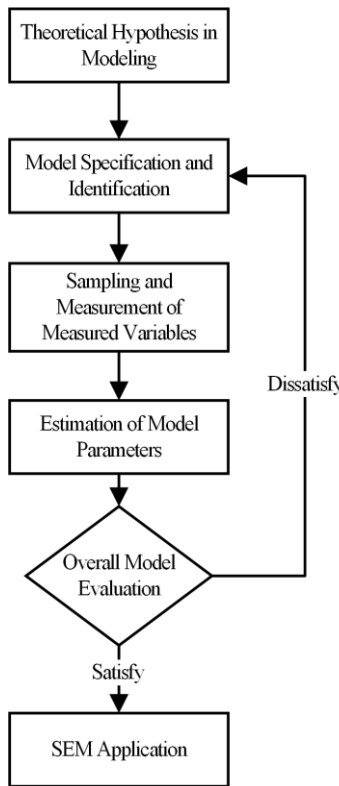

**Figure 1.** SEM calculation procedure.

*2.2. Calculation of the Contribution of Affecting Factors*

Assuming that *i* refers to the latent variable that directly affects reclaimed water utilization, and *j* is the latent variable that indirectly affects reclaimed water use through latent variable *i*, the formula for measuring the contribution of latent variables to reclaimed water use in the SEM is:

$$P_{ij} = \frac{|X_i Y_{ij}|}{\sum_{i=1, j=1} |X_i Y_{ij}|} \times 100\% \tag{4}$$

where $P_{ij}$ denotes the contribution of latent variables to reclaimed water use; $X_i$ refers to the coefficient of the direct impacts on reclaimed water utilization; $Y_{ij}$ is the coefficient of the direct impacts on latent variable *i* through latent variable *j*; $X_i Y_{ij}$ represents the coefficient of the impacts of latent variable *j* on reclaimed water utilization through latent variable *i*.

**3. Research Design**

The prediction of the potential availability of reclaimed water involves three steps. The first step is to establish an SEM. A structural model and a measurement model are established for the influencing factors of reclaimed water development and utilization, so that the SEM can well explain the changes in the utilization volume of reclaimed water. The second step is to analyze the contribution of indicators in the SEM to the development

and utilization of reclaimed water, and thus the critical indicators for reclaimed water development and utilization can be identified. The third step is to establish an REM. A regression analysis of the identified key indicators and the volume of reclaimed water development and utilization is performed to obtain a regression equation and thereby reach the goal of predicting the volume of reclaimed water development and utilization in China in 2025. Figure 2 shows the specific steps to predict the potential availability of reclaimed water development and utilization.

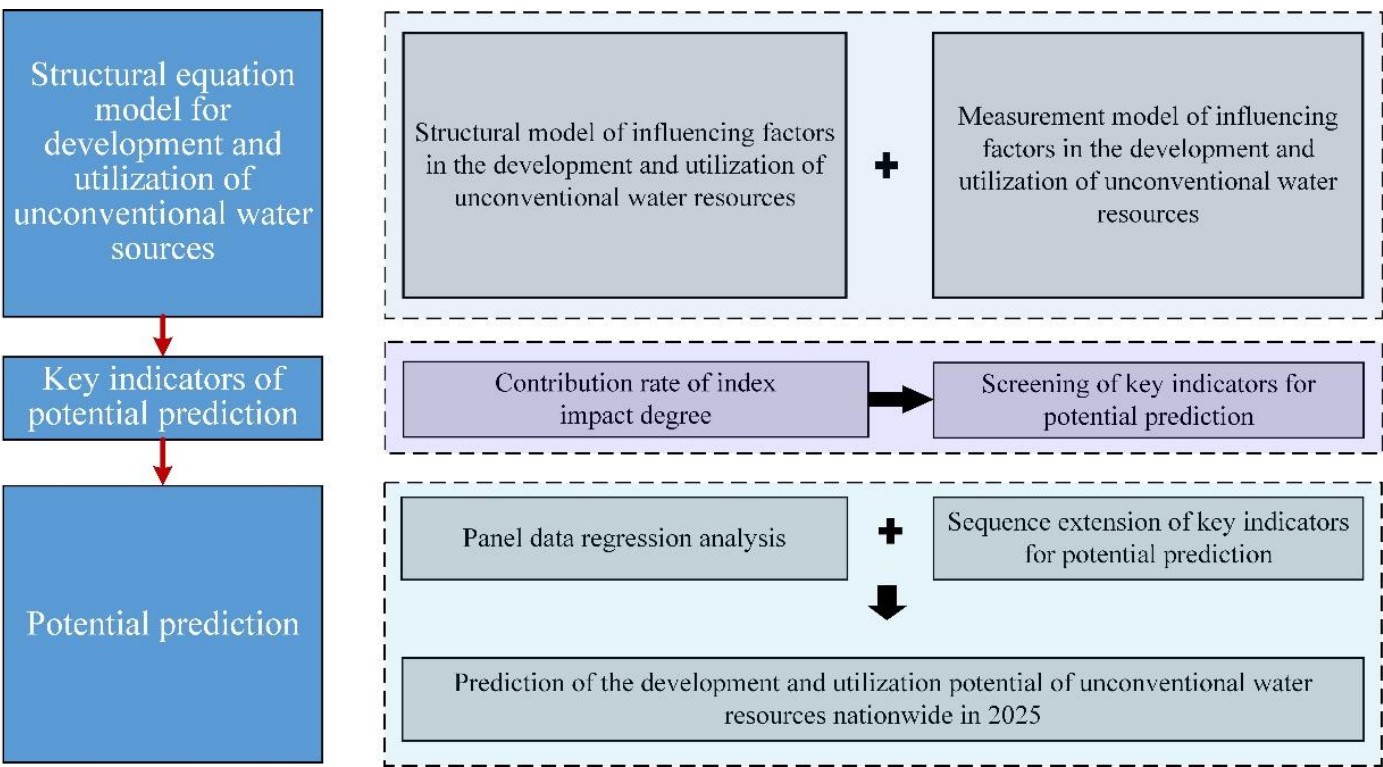

**Figure 2.** Workflow of prediction of the potential availability of reclaimed water.

## 4. Results and Discussions

### 4.1. Structural Model of the Influencing Factors in the Development and Utilization of Reclaimed Water

As is known, the use of reclaimed water is affected by both drivers, including characteristics of reclaimed water, regional water scarcity, climate change, and water use intensity, and restraints, such as engineering projects, economic cost, technological strength, and public acceptance. Hence, this paper explored the factors affecting the potential availability of reclaimed water from the perspectives of drivers and restraints. Built on supply and demand balancing, the study established a non-recursive structural mechanism framework for reclaimed water utilization, having taken into consideration such elements as regional water resources endowment, characteristics of water resources, scarcity of conventional water resources, ecosystem vulnerability, utilization levels, economic and social conditions, and the sufficiency of auxiliary engineering projects.

#### 4.1.1. Non-Recursive Structural Mechanism for Water Supply and Demand

Reclaimed water use is coordinated and driven by supply and demand. Supply-driven water utilization involves the production and utilization of reclaimed water by suppliers, specifically referring to the supply of reclaimed water sources, production facilities, human resources, and incentive policies. On the other hand, the demand-driven approach serves the interest of the users of reclaimed water, which involves industrial reuse, supply to the environment, miscellaneous uses in urban areas, and agricultural irrigation. As supply and

demand constitute the supply chain of reclaimed water use, their interconnection, along with the reciprocal relationship between subsystems and their feedback loop, presents itself. This paper, therefore, divided driving factors into supply and demand factors, and built a recursive structural model to indicate the dynamic relationship between supply-based and demand-driven systems.

4.1.2. Mechanism for Supply Restraints

Restraints on the reclaimed water supply merely reflect the impacts of restraining factors on the supply of reclaimed water. To be specific, they refer to strict quota or access restrictions posed by non-market-based restraints on the amount, prices, or actors of the reclaimed water supply. They will drive the cost of the reclaimed water supply, resulting in a less efficient supply.

Restraints on the reclaimed water supply can be exerted in direct and indirect ways. The former involves an unsound system for administering the exploration and use of reclaimed water and a lag in the development of incentive policies. In contrast, the latter includes the high costs of reclaimed water treatment and financing as a result of an unimproved network of auxiliary engineering projects on reclaimed water and outdated water treatment techniques. With that, this paper described a pathway to show how the utilization of reclaimed water is affected by supply restraints.

4.1.3. Major Factors of the Restraining Mechanism

According to existing studies, the major restraints that reflect the reality of reclaimed water utilization are as follows:

(1) Engineering [22]: the availability of reclaimed water is limited by water source characteristics and existing water conservancy facilities. That means stepped-up efforts to improve the facilities in plants for reclaimed water or wastewater treatment and the network of auxiliary pipelines hold the key to motivating reclaimed water use.

(2) Economic conditions [23]: projects on utilizing reclaimed water are a public investment from the perspective of the structure of the national economy, and factors such as local economic growth, treatment facilities, and the cost of construction should be taken into account.

(3) Techniques [24]: the progress of water quality treatment techniques is of great importance to reclaimed water application. The advanced treatment technology that reclaimed water plants boast has evolved from biological aerated filtering in the 1980s to today's hyperfiltration (reverse osmosis), superconducting magnetic separation, and membrane bioreactor. As a result, the efficiency of reclaimed water treatment has been increased considerably.

(4) Water environment [25]: there are certain impacts of reclaimed water use on the current water environment, specifically on the quantity and quality of existing water resources. For water quantity, reclaimed water, dubbed the second water source in urban areas, is water upon advanced treatment by sewage treatment plants before being supplied to downstream users. With reclaimed water utilization scaling up, the amount of water available in the natural water cycle will decrease, consequently creating negative impacts on the water environment. Regarding water quality, the standards of replenishing the natural environment with reclaimed water improved as China recommits its endeavors. The move helps make a better water environment, but given the weak spots in risk control of reclaimed water utilization, unreasonable behaviors still pose risks of damaging the water environment.

Given the analysis of the abovementioned supply–demand non-recursive structural mechanism and the mechanism for supply restraints, and considering the restraints on the engineering, economic, technical, and environmental fronts, we constructed a framework of non-recursive structural mechanisms for the factors affecting reclaimed water utilization, as is shown in Figure 3.

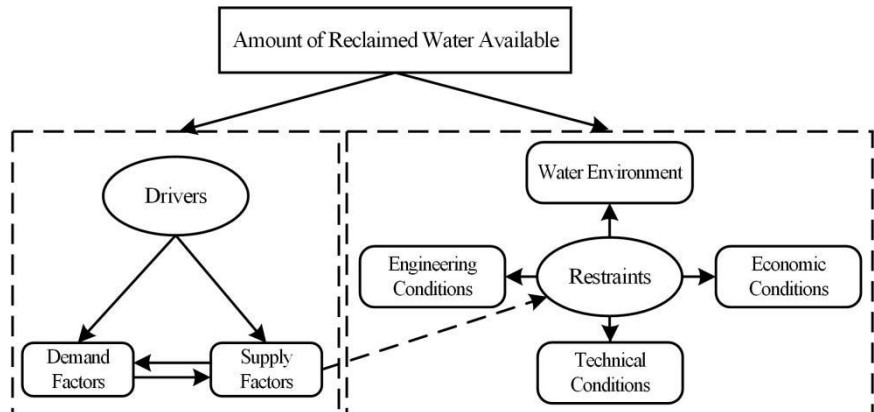

**Figure 3.** Framework of non-recursive structural mechanisms for factors affecting reclaimed water utilization.

*4.2. Measurement Model of the Influencing Factors for the Development and Utilization of Reclaimed Water*

4.2.1. Selection of the Measured Variables of the Driving Factors

(1) Supply factors: to study the impacts of supply factors on reclaimed water utilization, we selected four indicators, namely, the total water resources, the average per capita water resources, the amount of wastewater treated, and the population of workers in water treatment. Among them, total water resources and average per capita water resources indicate the water resources endowment, which reflects the potential replenishment of local water resources to reclaimed water sources. Since the water that reclaimed water plants process is the tailwater upon treatment of sewage treatment plants, the amount of sewage treated is highly relevant to the stability of reclaimed water supply. Finally, the population of water treatment workers suggests how market forces and relevant policies drive the reclaimed water industry.

(2) Demand factors: increasing the availability of reclaimed water holds the key to basing urban development on the local water system. That means the utilization of reclaimed water is intertwined with urbanization, which is a sure way towards new-type urbanization. Therefore, we selected total urban area and urban population density as demand factors for this study as they reveal the progress of urbanization and the demand for reclaimed water use in this process. In China, the amount of reclaimed water used in each field remains unclear. Studies, however, have found that reclaimed water has an increasingly important role in business operations, public services, people's daily life, and the environment, and the correlation between the consumption of reclaimed water and water use in each area increases progressively every year. Hence, this paper also considered the indicator of water utilization on each front to drive the demand for reclaimed water use.

4.2.2. Selection of the Measured Variables of the Restraints

(1) Engineering: having considered such factors as the quantity and quality of reclaimed water, and the reality of the auxiliary pipeline network, we identified seven indicators—the daily capacity of urban wastewater treatment, density of water supply pipelines in built-up areas, capacity of plants with grades II and III wastewater treatment technology, wastewater treatment rate, number of wastewater treatment plants, and production capacity of water supply—as the measured variables of engineering conditions that restrain reclaimed water use.

(2) Economic conditions: the study regarded five indicators, namely, the gross regional product (GRP), per capita GRP index, investment in industrial wastewater management, investment in wastewater treatment facilities, and investment in reclaimed water treatment facilities, as the measured variables of economic conditions that restrain reclaimed water use.

(3) Technical strength: considering treatment techniques and the adoption of technical measures, among others, we deemed the number of processed applications for water treatment patents as the measured variable of technical strength restraining reclaimed water use.

(4) Water environment: this paper considered the average annual precipitation, average annual evaporation, groundwater resources, underground water resources, proportion of rivers with a rating lower than grade V, and water use intensity as the measured variables of the water environment that restrain reclaimed water use.

Founded on the aforementioned analysis of factors driving and restraining reclaimed water use and combined with the non-recursive structural framework for affecting factor mechanisms, we considered the directly unmeasurable variables of drivers and restraints as level I latent variables in the structural equation model, and the other six directly unmeasurable variables, such as supply and demand factors and engineering conditions, as level II latent variables. Then, we selected measured variables based on each level II latent variable for a measurement model (Table 1). This, coupled with the non-recursive structural framework, constituted an improved SEM for reclaimed water use.

**Table 1.** Measurement model for reclaimed water use.

| Level I Latent Variables | Level II Latent Variables | Measured Variables | No. | Unit | Source |
|---|---|---|---|---|---|
| Drivers | Supply Factors | Total Water Resources | E1 | 100 Mm$^3$ | [26] |
| | | Average Per Capita Water Resources | E2 | m$^3$/capita | [26] |
| | | Quantity of Wastewater Treated | E3 | 100 Mm$^3$ | [27] |
| | | Population of Water Treatment Workers | E4 | people | [28] |
| | Demand Factors | Total Urban Area | E5 | km$^2$ | [27] |
| | | Density of Urban Population | E6 | people/km$^2$ | [27] |
| | | Total Amount of Water Consumption | E7 | 100 Mm$^3$ | [26] |
| | | Water Consumption in Agriculture | E8 | 100 Mm$^3$ | [26] |
| | | Water Consumption in Industry | E9 | 100 Mm$^3$ | [26] |
| | | Water Consumption in Daily Life | E10 | 100 Mm$^3$ | [26] |
| | | Water Consumption in Ecology | E11 | 100 Mm$^3$ | [26] |
| | | Per Capita Water Consumption | E12 | L/capita | [26] |
| | | Water Consumption in Production | E13 | 100 Mm$^3$ | [26] |
| Restraints | Engineering Conditions | Daily Capacity of Urban Wastewater Treatment | E14 | 10,000 m$^3$ | [27] |
| | | Density of Water Supply Pipelines in Built-up Areas | E15 | km/km$^2$ | [27] |
| | | Density of Drainage Pipelines in Built-up Areas | E16 | km/km$^2$ | [27] |

| Level I Latent Variables | Level II Latent Variables | Measured Variables | No. | Unit | Source |
|---|---|---|---|---|---|
| | | Capacity of Plants with Grade II and III Wastewater Treatmen Technology | E17 | 10,000 m$^3$ | [27] |
| | | Wastewater Treatment Rate | E18 | % | [27] |
| | | Wastewater Treatment Plants | E19 | — | [27] |
| | | Production Capacity of Water Supply | E20 | 10,000 m$^3$/d | [28] |
| | Economic Conditions | GRP | E21 | 100 M RMB | [28] |
| | | Per Capita GRP Index | E22 | — | [28] |
| | | Investment in Industrial Wastewater Management | E23 | 10,000 RMB | [28] |
| | | Investment in Wastewater Treatment Facilities | E24 | 10,000 RMB | [27] |
| | | Investment in Reclaimed Water Treatment Facilities | E25 | 10,000 RMB | [27] |
| | Technical Strength | Number of Processed Applications for Water Treatment Patents | E26 | pcs | [28] |
| | Water Environment | Average Annual Precipitation | E27 | mm | [26] |
| | | Average Annual Evaporation | E28 | mm | [26] |
| | | Groundwater Resources | E29 | 100 Mm$^3$ | [26] |
| | | Underground Water Resources | E30 | 100 Mm$^3$ | [26] |
| | | Proportion of Rivers with a Rating Lower than Grade V | E31 | % | [26] |
| | | Water Use Intensity | E32 | % | [27] |

### 4.3. Screening of Key Indicators for Potential Prediction

To screen the key indicators for potential prediction, we used the data on reclaimed water use in provinces (autonomous regions and municipalities directly under the central government) from 2015 to 2017 organized by the National Office of Water Conservation in 2017 and the statistics on reclaimed water use across China from 2011 through 2018 [26], as well as the panel data of 32 indicators in the measurement model [26–28], as essential data and imported them into AMOS software after data standardization. This, together with the estimation of parameters in the established measurement model for reclaimed water use through maximum likelihood analysis, enabled us to measure the factor loading of the model.

#### 4.3.1. SEM Model Fitting Effect

To achieve a sound fitting effect for the SEM model, the study selected several standard indices for evaluating SEMs, including the chi-square test, goodness of fit index, alternative index, and residual analysis (See Table 2).

**Table 2.** Fitting effect evaluation indicators for the structural equation model.

| Result Type | Chi-Square Test | Goodness of Fit Index | | Alternative Index | | Residual Analysis |
|---|---|---|---|---|---|---|
| | $\chi^2/df$ | GFI | NFI | CFI | RMSEA | SRMR |
| Standard Value | 1~3 | >0.90 | >0.90 | >0.95 | <0.08 | <0.08 |
| Fitting Value | 1.236 | 0.914 | 0.902 | 0.987 | 0.063 | 0.031 |

As Table 2 shows, the fitting value of the chi-square test $\chi^2/df$ of the model is in the reasonable range, suggesting that the model has good fitting performance and good interpretability, not subject to the problem of overfitting [29]. Both the goodness of fit index and the alternative index meet the standard, which indicates a relatively large proportion of explanatory measured data variance to covariance and a high goodness of fit [30]. In a residual analysis, the SRMR value failed to reach the critical threshold, meaning small residuals in the model overall [31]. On the whole, the results showed that the non-recursive structural equation model is up to test standards and with good fitting effects.

### 4.3.2. Contribution of Indicator Impacts

In SEM, correlations between variables are linked by the structural parameters of the model. The correlation between latent variables is expressed by the parameter $\beta$, and its value, represented by factor loading, reflects the correlation extent; the squared factor loading is the percent of variance in that indicator variable explained by the factor. The correlation between latent and measured variables is expressed by $\lambda$, which shares a similar meaning to the parameter $\beta$. The symbol $\varphi$ represents the covariance matrix of the error term among latent variables, indicating the intensity of correlation between measured residuals. Moreover, $e$ in this study was used to suggest the possible measured errors of variables. Table 3 shows the correlations among factors affecting reclaimed water use and their respective intensity.

**Table 3.** Influencing paths of factors affecting reclaimed water use in the structural equation model.

| Serial Number | Variables | Application Direction | Variables | Standardized Regression Weights | Serial Number | Variables | Application Direction | Variables | Standardized Regression Weights |
|---|---|---|---|---|---|---|---|---|---|
| 1 | Drivers | ← | Reclaimed Water Use | 0.79 | 23 | E12 | ← | Demand Factors | 0.01 |
| 2 | Restraints | ← | Reclaimed Water Use | 0.36 | 24 | E13 | ← | Demand Factors | 0.10 |
| 3 | Supply Factors | ← | Drivers | 0.84 | 25 | E14 | ← | Engineering Conditions | 0.97 |
| 4 | Demand Factors | ← | Drivers | 0.11 | 26 | E15 | ← | Engineering Conditions | 0.26 |
| 5 | Restraints | ← | Supply Factors | 0.25 | 27 | E16 | ← | Engineering Conditions | 0.19 |
| 6 | Engineering Conditions | ← | Restraints | 0.96 | 28 | E17 | ← | Engineering Conditions | 0.94 |
| 7 | Economic Conditions | ← | Restraints | 0.87 | 29 | E18 | ← | Engineering Conditions | 0.31 |
| 8 | Technical Strength | ← | Restraints | 0.88 | 30 | E19 | ← | Engineering Conditions | 0.95 |
| 9 | Water Environment | ← | Restraints | −0.03 | 31 | E20 | ← | Engineering Conditions | 0.93 |
| 10 | Demand Factors | ← | Supply Factors | −0.14 | 32 | E21 | ← | Economic Conditions | 0.95 |
| 11 | Supply Factors | ← | Demand Factors | 0.23 | 33 | E22 | ← | Economic Conditions | −0.04 |
| 12 | E1 | ← | Supply Factors | 0.20 | 34 | E23 | ← | Economic Conditions | 0.68 |

**Table 3.** *Cont.*

| Serial Number | Variables | Application Direction | Variables | Standardized Regression Weights | Serial Number | Variables | Application Direction | Variables | Standardized Regression Weights |
|---|---|---|---|---|---|---|---|---|---|
| 13 | E2 | ← | Supply Factors | −0.18 | 35 | E24 | ← | Economic Conditions | 0.42 |
| 14 | E3 | ← | Supply Factors | 0.95 | 36 | E25 | ← | Economic Conditions | −0.02 |
| 15 | E4 | ← | Supply Factors | 0.63 | 37 | E26 | ← | Technical Strength | 0.96 |
| 16 | E5 | ← | Demand Factors | −0.01 | 38 | E27 | ← | Water Environment | 0.40 |
| 17 | E6 | ← | Demand Factors | 0.00 | 39 | E28 | ← | Water Environment | 0.81 |
| 18 | E7 | ← | Demand Factors | 0.95 | 40 | E29 | ← | Water Environment | −0.17 |
| 19 | E8 | ← | Demand Factors | 0.93 | 41 | E30 | ← | Water Environment | 0.20 |
| 20 | E9 | ← | Demand Factors | 0.88 | 42 | E31 | ← | Water Environment | 0.90 |
| 21 | E10 | ← | Demand Factors | 0.73 | 43 | E32 | ← | Water Environment | −0.16 |
| 22 | E11 | ← | Demand Factors | 0.26 | | | | | |

As Table 3 shows, the loading of the latent variables of the driving factors of reclaimed water use was 0.79, which means the latent variables of the driving factors reached an explanation level of 62.4% for the utilization of reclaimed water. The factor loading of the latent variables of the restraints was 0.36, meaning the explanation level of the reclaimed water use by such variables stood at 13%. The remaining 24.6% was considered an unexplainable estimation error, probably attributed to the measured error in variable measurement or to other latent variables.

As the error coefficients of the correlation between the estimation error and latent variables of the drivers, and between the estimation error and latent variables of the restraints ($\varphi_1 = 0.48$, $\varphi_2 = 0.62$) were small and their correlations were weak, we believed that there might exist other latent variables that could be used to explain reclaimed water use.

Upon analysis of the factor loading of the level II latent variables based on the latent variable of the restraints, it was found that the latent variable of the restraints was greatly affected by engineering conditions, economic reality, and technical strength, whose factor loadings were all above 0.85. On the contrary, the impact of the water environment latent variable on the latent variable of the restraints was relatively small ($\beta = -0.03$). As a result, we concluded that the water environment has a small restraining impact on reclaimed water use, while the restraining impacts of auxiliary engineering construction and the cost of wastewater treatment are more significant.

Having examined the structural model describing the correlation between the latent variable of the drivers and supply-side latent variables, and between the latent variables of the drivers and demand-side latent variables, and compared the correlation intensity between latent variables, we found that the latent variables of the supply factors had a more substantial impact on those of the drivers ($\beta = 0.84$), with an explanation level of 70.6%. Meantime, the analysis of the correlation between supply-side and demand-side latent variables showed that the factor loading of demand-side latent variables to supply-side ones was negative ($\beta = -0.14$), meaning the demand-side latent variables had a negligible impact on the supply-side variables, but were more affected by the latter. Hence, we think that the supply capacity of reclaimed water acts as a dominant factor that significantly drives reclaimed water utilization.

To further identify the influence of supply-demand latent variables, as well as the latent variables of engineering conditions, economic conditions, technical strength, and water environment on reclaimed water use, this paper, combined with the intensity of direct and indirect correlations between latent variables, quantitatively measured the contribution of each latent variable to the impacts of reclaimed water use (Table 4).

**Table 4.** Contribution of each latent and measured variable to the impacts on reclaimed water use.

| Level I Latent Variables | Contribution (%) | Level II Latent Variables | Contribution (%) | Measured Variables | Contribution (%) |
|---|---|---|---|---|---|
| Drivers | 65.6% | Supply Factors | 60.4% | Total Water Resources | 5.9% |
| | | | | Average Per Capita Water Resources | 4.4% |
| | | | | Quantity of Wastewater Treated | 45.2% |
| | | | | Population of Water Treatment Workers | 4.9% |
| | | Demand Factors | 5.2% | Total Urban Area | 0.1% |
| | | | | Density of Urban Population | 0.1% |
| | | | | Total Amount of Water Consumption | 0.8% |
| | | | | Water Consumption in Agriculture | 0.9% |
| | | | | Water Consumption in Industry | 1.2% |
| | | | | Water Consumption in Daily Life | 0.7% |
| | | | | Water Consumption in Ecology | 0.8% |
| | | | | Per Capita Water Consumption | 0.1% |
| | | | | Water Consumption in Production | 0.5% |
| Restraints | 34.4% | Engineering Conditions | 11.8% | Daily Capacity of Urban Wastewater Treatment | 1.2% |
| | | | | Density of Water Supply Pipelines in Built-up Areas | 7.6% |
| | | | | Density of Drainage Pipelines in Built-up Areas | 0.4% |
| | | | | Capacity of Plants with Grade II and III Wastewater Treatment Technology | 1.1% |
| | | | | Wastewater Treatment Rate | 0.7% |
| | | | | Wastewater Treatment Plants | 0.3% |
| | | | | Production Capacity of Water Supply | 0.5% |
| | | Economic Conditions | 9.6% | GRP | 0.1% |
| | | | | Per Capita GRP Index | 0.2% |
| | | | | Investment in Industrial Wastewater Management | 1.3% |
| | | | | Investment in Wastewater Treatment Facilities | 1.7% |
| | | | | Investment in Reclaimed Water Treatment Facilities | 6.3% |
| | | Technical Strength | 9.7% | Number of Processed Applications for Water Treatment Patents | 9.7% |
| | | Water Environment | 3.3% | Average Annual Precipitation | 0.5% |
| | | | | Average Annual Evaporation | 1.0% |
| | | | | Groundwater Resources | 0.2% |
| | | | | Underground Water Resources | 0.3% |
| | | | | Proportion of Rivers with a Rating Lower than Grade V | 1.1% |
| | | | | Water Use Intensity | 0.2% |

Table 4 shows that the impacts of the latent variables of the drivers on reclaimed water use were twice as significant as those of restraints. This means that the reclaimed water utilization has a great potential as it boasts more driving factors than restraining ones. Given supply-side restraints in the structural model, the latent variables of the supply factors exerted direct and indirect impacts on reclaimed water use, with a total contribution rate reaching 60.4%. The contribution of the latent variables of the demand factors to the impacts on reclaimed water use was merely 5.2%, suggesting that the supply capacity of reclaimed water is more robust than its demand due to the unknown risk of its use. Upon analyzing the contribution that the latent variables of the restraints made to the impacts on reclaimed water use, we found that a lack of improved engineering conditions, including the auxiliary pipeline network of reclaimed water, served as a dominant restraining factor, with a contribution rate of 11.8%. The secondary restraining factors included the regional economic conditions and technical strength in wastewater treatment, which accounted for 10% of the impacts on reclaimed water use, and the figure was merely 3.3% for the latent variable of the water environment.

### 4.3.3. Key Indicators for Potential Prediction

The analysis of the contribution of measured variable impacts showed that the impacts of four measured variables—the quantity of wastewater treated (E3), the density of water supply pipelines in built-up areas (E15), investment in reclaimed water treatment facilities (E25), and the number of processed applications for water treatment patents (E26), were greater than those of other measured variables, indicating a closer correlation with reclaimed water use. Considering the time series length of indicators and data accessibility, the study selected the above four variables as key indicators for predicting the potential of reclaimed water use.

(1)    The quantity of wastewater treated refers to the actual amount of wastewater processed by wastewater treatment plants (or facilities), including the amount treated by physical, biological, and chemical means.

(2)    The density of water supply pipelines in built-up areas shows how water supply pipelines are distributed in built-up areas. The calculation equation is as follows.

$$\text{Density of water supply pipelines in built-up areas} \ = \ \frac{\text{Total length of water supply pipelines}}{\text{Total area of built-up areas}} \tag{5}$$

(3)    Investment in reclaimed water treatment facilities refers to reclaimed water treatment projects with a planned investment of more than RMB 50,000 each, which include the renovation, rebuilding, extension, and establishment of fixed assets.

(4)    The number of processed applications for water treatment patents shows the number of processed patent applications in the treatment of water, wastewater, sewage, and sludge.

### 4.4. Panel Data Regression Analysis Results

Built on the screening of the aforementioned key indicators for potential prediction, a regression model involving the indicators and reclaimed water consumption was constructed, with an aim to predict the potential quantity of reclaimed water.

### 4.4.1. Unit Root Test

Analyzing panel data requires the unit root test on the variable sequence of the model, with an aim to examine its residual stability and avoid spurious regression. The unit root test, in this paper, was based on the augmented Dickey–Fuller (ADF) test [32] and the Kwiatkowski–Phillips–Schmidt–Shin (KPSS) test [33]. Table 5 presents the test results.

Table 5 showed that E25 rejects the null hypothesis of a unit root, but after the first-order difference operation, each variable rejects the original hypothesis. That means after the first-order difference operation, variables have no unit root and enjoy a stable sequence. Therefore, original variables can be imported into the regression model.

**Table 5.** Results [1] of the unit root test on panel data.

| Variables | ADF Test | KPSS Test | Stability | First-Order Difference | Stability |
|-----------|----------|-----------|-----------|------------------------|-----------|
| E3 | −3.84 (0.02) | 0.051 (>0.1) | Stable | — | — |
| E15 | −4.59 (<0.01) | 0.203 (>0.1) | Stable | — | — |
| E25 | −3.15 (0.10) | 0.068 (>0.1) | Unstable | 0.035 (>0.1) | Stable |
| E26 | −4.43 (<0.01) | 0.285 (>0.1) | Stable | — | — |

[1] The $p$-value is in brackets.

### 4.4.2. REM for Potential Prediction

Based on the stationary test of panel data variables, common panel data regression models were established: fixed-effects model (FEM), random-effects model (REM) and mixed-effects model (MEM). The least-squares method was then employed to perform parameter estimation of these three models. Table 6 shows the results of parameter estimation.

**Table 6.** Estimation results [1] on parameters in different models.

| Coefficient | FEM | REM | MEM |
|-------------|-----|-----|-----|
| Intercept | — | 14,080.9 (0.009) | 14,982 (0.02) |
| E3 | 0.443 (0.129) | −0.517 (0.007) | −0.356 (0.000) |
| E15 | −223.6 (0.674) | −1706.4 (0.000) | −1819.6 (0.000) |
| E25 | −0.002 (0.886) | 0.052 (0.000) | 0.066 (0.000) |
| E26 | 0.161 (0.236) | 0.209 (0.000) | 0.239 (0.019) |

[1] The $p$-value is in brackets.

Given the selection of regression models, the F statistic and the Hausman statistic were constructed, with the former used to test the MEM and FEM models and the latter the FEM and REM models. Results (See Table 7) of the two tests showed that the F value is greater than the confidence level threshold and the $p$-value is less than 0.001. This indicates the rejection of the original hypothesis and the significance of the fixed effects. The Hausman test found that the Hausman statistic was 2.718 and its corresponding probability was 0.746. This suggests that the test accepts the original hypothesis at the confidence level of 5%, meaning the REM is accepted. Therefore, we used the REM for the empirical analysis.

**Table 7.** Statistic test results of models.

| Tests | Statistic | $p$-Value |
|-------|-----------|-----------|
| F Test | 15.043 | <0.001 |
| Hausman Test | 2.718 | 0.746 |

Regression models for panel data are sensitive to extreme values, and multicollinearity among independent variables and parametric heteroscedasticity could result in regression errors, which lead to uncertainty about the regression coefficients. As a result, regression models cannot show the impacts of the explanatory variables on the response variables, as well as the changes of such impacts. Therefore, in light of the model stability, we conducted a quantile regression on the parameters to better understand the parametric uncertainty and the changes of the impacts that the explanatory variables have on the

response variables. The estimation results of each REM parameter at 10–90% quantiles are shown in Figure 4.

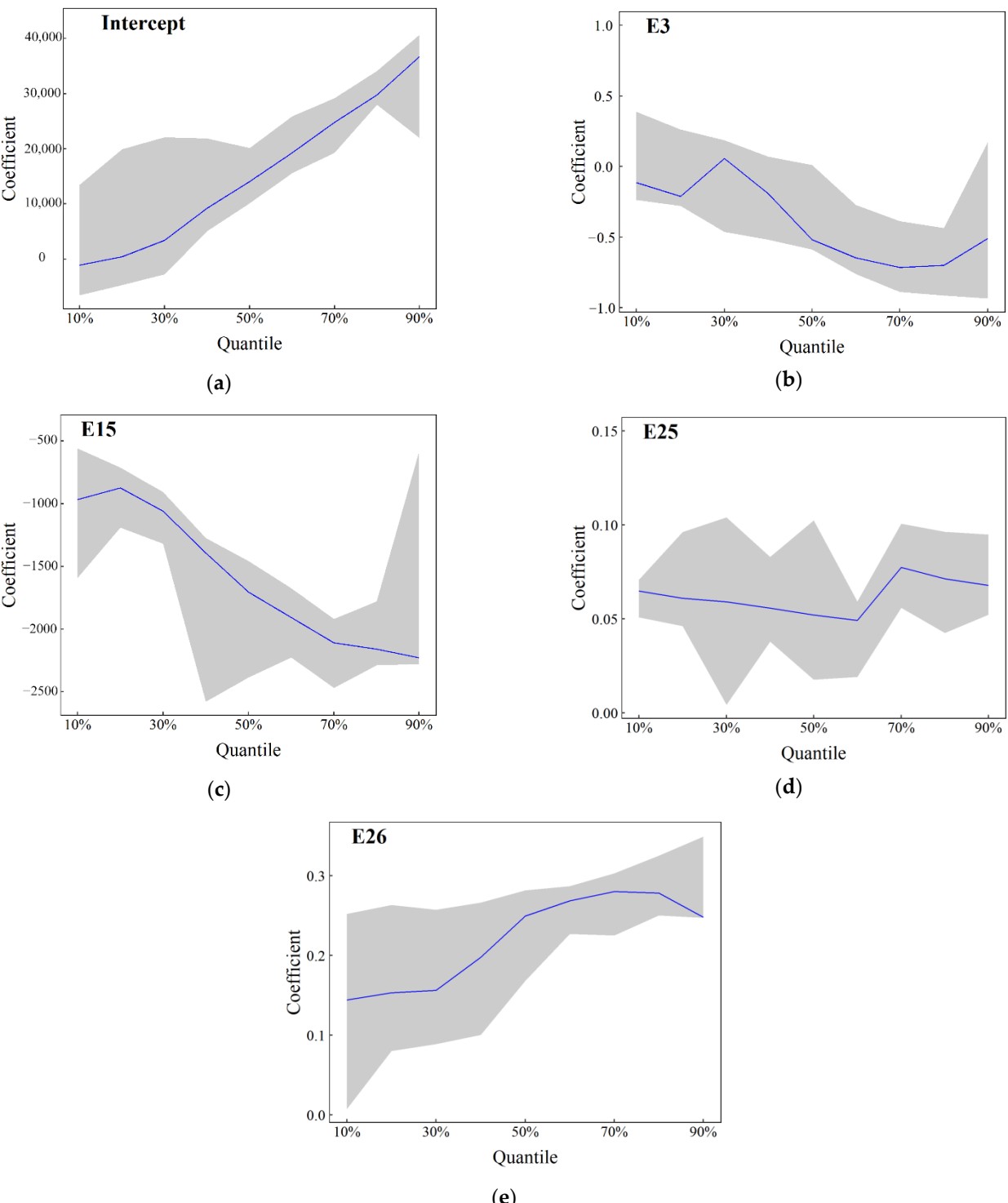

**Figure 4.** REM parameters at 10–90% quantiles. (**a**) Coefficient of intercept at 10–90% quantile. (**b**) Coefficient of E3 at 10–90% quantile. (**c**) Coefficient of E15 at 10–90% quantile. (**d**) Coefficient of E25 at 10–90% quantile. (**e**) Coefficient of E26 at 10–90% quantile.

According to the analysis of Figure 4, the regression coefficients of E25 and E26 at each quantile are both positive, which means that there is a positive relationship between the indicator and the development and utilization of reclaimed water. It is consistent

with indicator implications, but they fluctuated largely at 0.1–0.3 quantiles. E3 and E15 negatively affect reclaimed water use, which also justifies indicator implications. The estimation symbols of the model parameters in different ranges of quantiles are in line with those of the REM parameters, indicating a steady estimation of the REM parameters.

### 4.5. Potential Prediction Results

4.5.1. Extension of Key Indicator Sequence for Potential Prediction

The calculation of future availability of reclaimed water based on the RME necessitates the extension of each key indicator sequence for the potential prediction. The number of processed applications for water treatment patents and the quantity of wastewater treated, among others, featured prominently over time, and their fitting curves are shown in Figures 5 and 6.

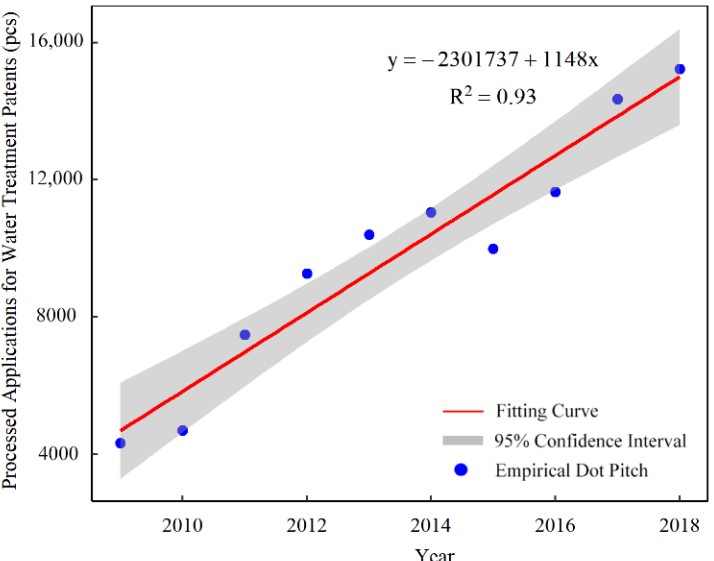

**Figure 5.** Investment in drainage fixed assets.

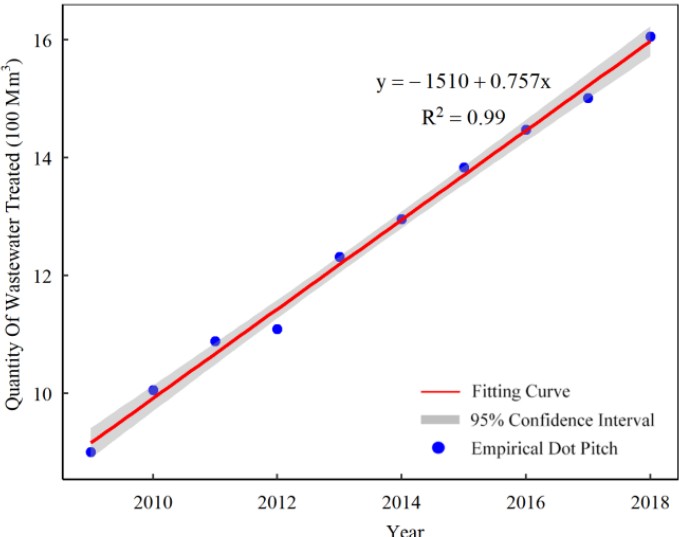

**Figure 6.** Investment in drainage fixed assets.

Figure 5 suggests that the number of processed applications for water treatment patents in China is rising year-on-year. For the foreseeable future, the fitting curve that relates the number of processed applications for water treatment patents to time variable is:

$$y = -2301737 + 1148x \tag{6}$$

where $y$ is the number of processed applications for water treatment patents; $x$ denotes a given year.

As is shown in Figure 6, economic and social development comes with an annual increase in the amount of wastewater treated, which has a significant linear relationship with the time variable. The relevant fitting curve is:

$$y = -1510 + 0.757x \tag{7}$$

where $y$ represents the amount of wastewater treated; $x$ denotes a given year.

According to investment in drainage (comprising sewage and sludge) facilities, specifically consisting of investment in sewage treatment, sludge disposal, reclaimed water use, and others (as shown in Figure 7), the ratio of investment in reclaimed water use sees a slight fluctuation, meaning that for years the fixed-asset investment in this aspect has been kept stable.

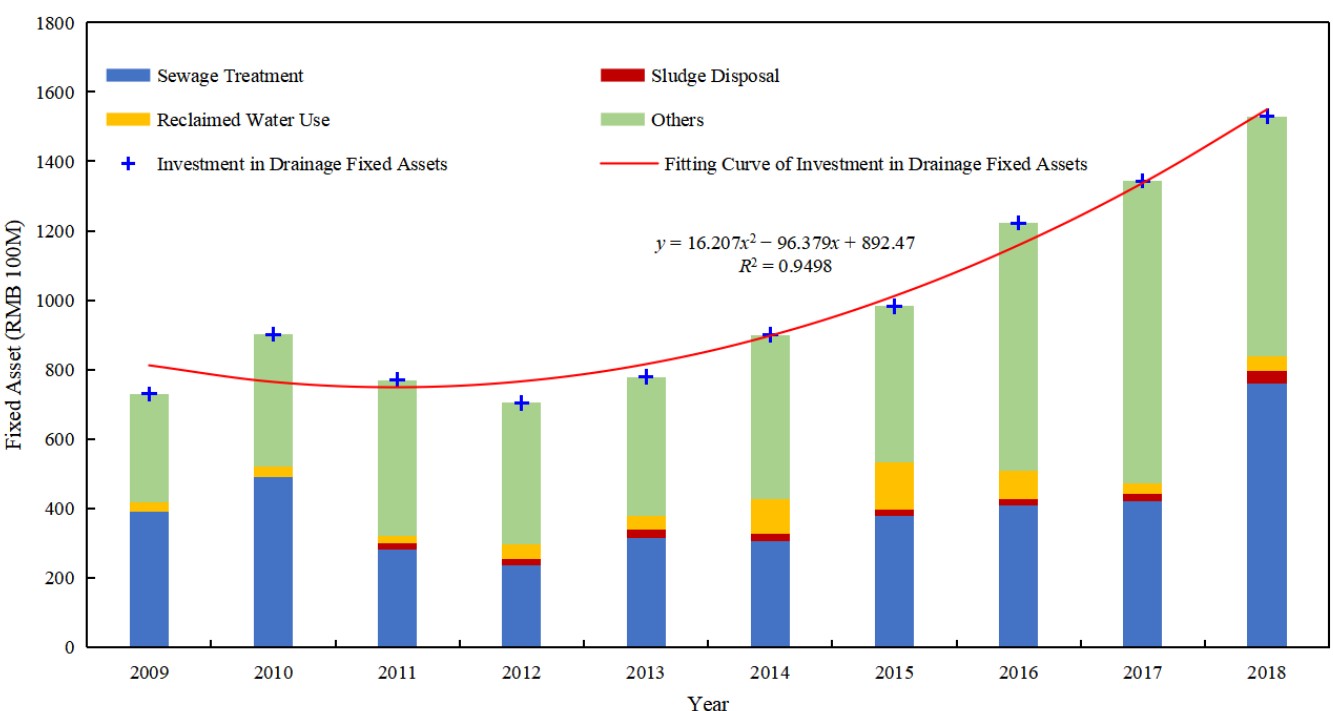

**Figure 7.** Investment in drainage fixed assets.

Figure 7 indicates a significant correlation between investment in drainage fixed assets and time variables, and for years, investment in facilities for reclaimed water use had accounted for 5.54% of the total investment in drainage fixed assets. Its time series length can be extended by shaping the fitting curve of investment in drainage facilities and combining the average proportion of investment in drainage fixed assets.

The fitting curve of investment in drainage fixed assets is:

$$y = 892.47 - 96.379x + 16.207x^2 \tag{8}$$

where $y$ is the investment in drainage fixed assets; $x$ denotes a given year.

It is known that the density of water supply pipelines and the total area of built-up areas are closely connected with the total length of water supply pipelines. The density can be measured by constructing the fitting curves (as shown in Figure 8) of the total area of built-up areas and the length of water supply pipelines over time before using Equation (5).

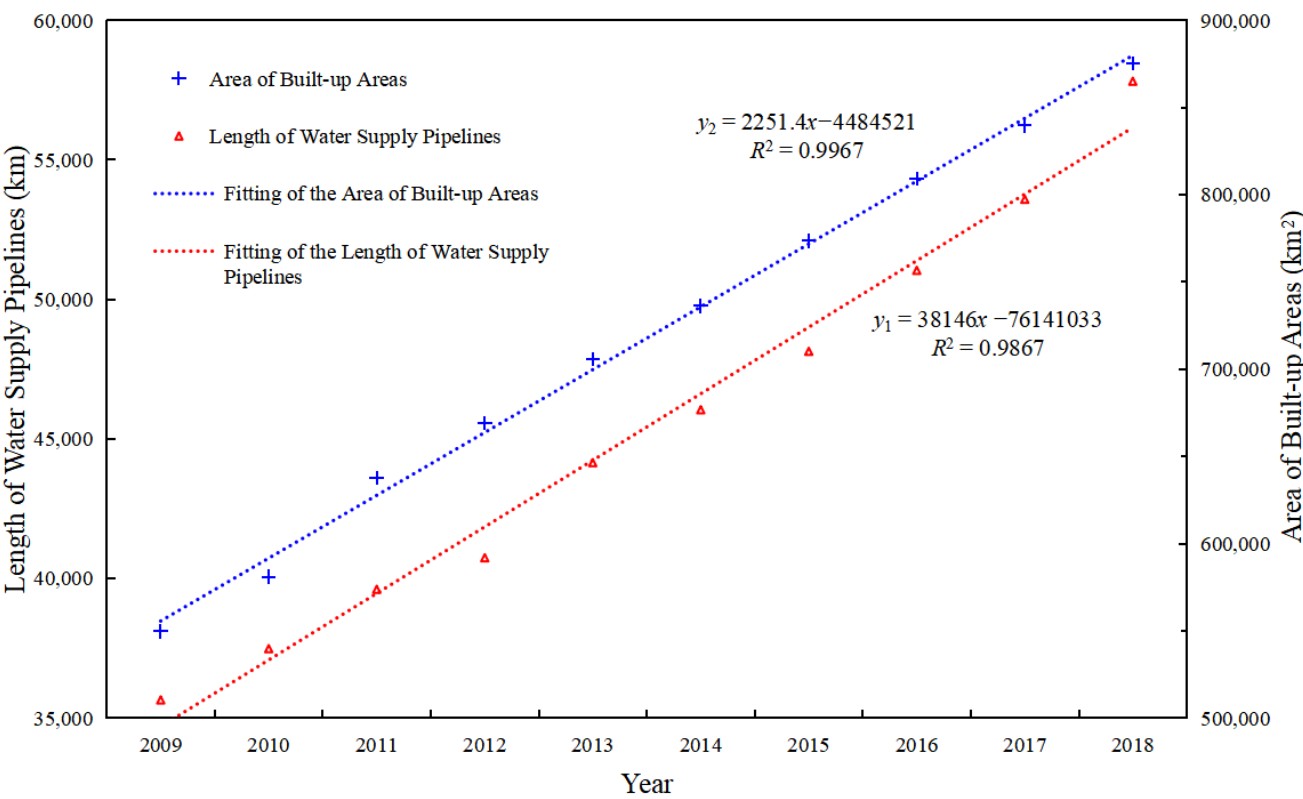

**Figure 8.** Fitting curves of the length of water supply pipelines and the total area of built-up areas.

The fitting curves of the length of water supply pipelines and the total area of built-up areas are:

$$y_1 = -76141033 + 38146.4x \tag{9}$$

$$y_2 = -4484521 + 2251.4x \tag{10}$$

where $y_1$ is the length of water supply pipelines; $y_2$ is the total area of built-up areas; $x$ represents a given year.

On the basis of the above fitting curves and all indicator implications, we extended the time series of each indicator to 2025, with estimation results displayed in Table 8.

**Table 8.** Estimation results of each key indicator for potential prediction.

| Year | E3 (100 Mm$^3$) | E15 (km/km$^2$) | E25 (RMB 100 M) | E26 (pcs) |
|---|---|---|---|---|
| 2019 | 16.7 | 14.7 | 99.4 | 16,075 |
| 2020 | 17.5 | 14.9 | 114.7 | 17,223 |
| 2021 | 18.2 | 15.1 | 131.9 | 18,371 |
| 2022 | 19.0 | 15.3 | 150.8 | 19,519 |
| 2023 | 19.8 | 15.6 | 171.5 | 20,667 |
| 2024 | 20.5 | 15.8 | 194.0 | 21,815 |
| 2025 | 21.3 | 16.0 | 218.3 | 22,963 |

### 4.5.2. Prediction on Potential Availability

Today, the amount of reclaimed water used in China stands at 20.137 million cubic meters per day. According to the established REM and the results on the sequence extension of each indicator for the potential prediction, we measured the potential availability of reclaimed water from 2019 through 2025 and prepared the greyscale map (Figure 9) of potential reclaimed water availability at 25%, 50%, and 75% quantiles.

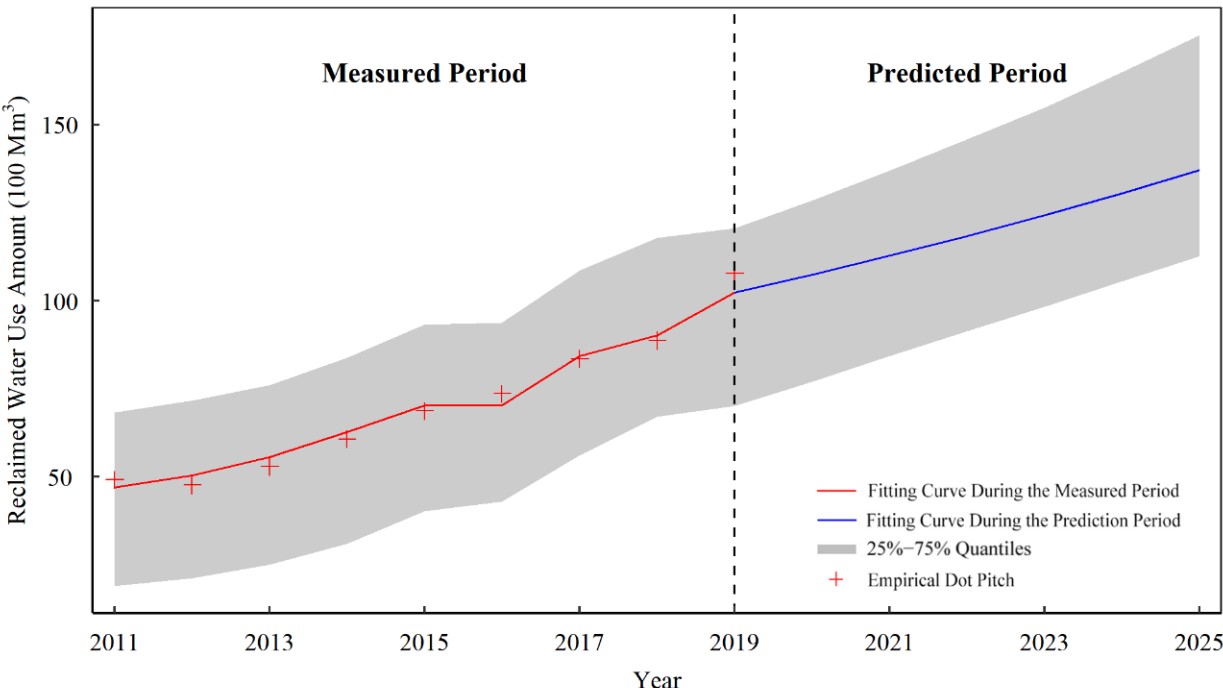

**Figure 9.** Potential reclaimed water availability.

Founded on the REM and the predicted values of each key indicator for potential prediction between 2019 and 2025, we expected the potential availability of reclaimed water in China to reach about 13.7 billion cubic meters, and the predicted use at 25–75% quantiles ranged from 11.3 to 17.6 billion cubic meters. A slow increase in potential availability at lower quantiles was related to a great change in the negative value of model parameters at lower quantiles.

### 4.5.3. Fitting Error Analysis

Upon comparative analysis between the predicted and measured values, it was found that the predicted and empirical dot pitch in the REM for the measured period boasted a high fitting precision (See Table 9), with the largest fitting error standing at 5.9%. Furthermore, the regression curves showed how the empirical dot pitch fluctuated. All these suggested that selecting four types of key indicators for the potential prediction through an RDA analysis can greatly represent the raw information about reclaimed water use. From the perspective of the developments in reclaimed water utilization, the effort to use reclaimed water is expected to maintain growth from 2019 through 2025. This is consistent with the planning for a better-structured urban water supply while balancing water resource utilization, industrial structure, and spatial distribution. The predicted results of the REM are, therefore, reasonable.

**Table 9.** Fitting results of the REM [1].

| Year | Measured Amount (100 Mm³) | Fitting Amount (100 Mm³) | Fitting Error (%) |
|------|---------------------------|--------------------------|-------------------|
| 2011 | 49.1 | 47.0 | −4.3 |
| 2012 | 47.6 | 50.4 | 5.9 |
| 2013 | 52.9 | 55.6 | 5.2 |
| 2014 | 60.7 | 62.7 | 3.3 |
| 2015 | 68.7 | 70.2 | 2.2 |
| 2016 | 73.6 | 70.2 | −4.6 |
| 2017 | 83.5 | 84.3 | 1.0 |
| 2018 | 88.7 | 90.2 | 1.7 |

[1] $R^2 = 0.847$.

*4.6. Model Results Discussions*

On researching the potential availability of reclaimed water, this paper integrated the SEM with the REM, which had improvement in three aspects, compared with the analysis of supply-demand balancing. First, conducting a supply-demand balancing analysis requires massive data of multifarious types. For instance, Tang [19] argued that to predict the potential availability of reclaimed water necessitates data on reclaimed water use in relevant sectors or fields, such as industry, urban greening, and lake landscaping. With the SEM-REM approach, dependence on these data is remarkably reduced, and the prediction can be done by selecting key indicators with a greater contribution to the impacts on reclaimed water use, based on the framework of mechanisms for affecting factors and existing local statistics before extending data. Hence, the method is particularly applicable to the areas lacking complete statistical data. Second, using a supply-demand balancing analysis to set up future scenarios involves many subjective thoughts of the decision makers. For example, Zhang [34] believed that the reclaimed water use ratio (the proportion of reclaimed water consumption to the sewage treated amount) is expected to surpass 80% by 2025. Chu [35] reckoned that in 2020, reclaimed water would account for 60% and 80% of the total water consumption for green belt sprinkling and street flushing, respectively, and the amount of reclaimed water would see an annual growth of 10% in water consumption in public lavatories. The setting of these figures, however, cannot hold water, nor can it reflect the progress on reclaimed water use in a science-based manner. With the proposed SEM-REM approach, the selection of key indicators for potential prediction is based on the mechanisms of factors affecting reclaimed water use, and the extension of indicator data is consistent with laws of mathematical statistics at the confidence level of 90%. Therefore, indicators selection and the statistical implications of data are less undermined by decision makers' subjective thoughts, rendering the measurement of reclaimed water availability objective and science based. Third, the utilization of reclaimed water involves multiple factors, such as the natural environment, the ecology, and economic and social realities [36]. Moreover, the interconnectivity between the utilization scale and drivers and restraints, as well as their response mechanisms, involves various latent variables that cannot be observed directly, such as the technical strength in reclaimed water treatment and regional economic growth. Although supply-demand balancing is based on the law of water balance and capable of analyzing the reality of reclaimed water use, it has apparent limitations when it comes to measuring reclaimed water availability for a future period. The reason is that despite an analysis of the impacts of factors that are measurable directly, such as groundwater resources and water consumption in production and ecology [37] by measuring the amounts of water supply and consumption, the method fails to quantitatively analyze the impacts of those factors that are unmeasurable directly, including economic and social development and technical strength in water treatment. As a matter of fact, grasping the growing trend in the regional economy, technology, and policy is key to predicting potential reclaimed water use. Built on the analysis of a structural equation model, the proposed SEM-REM method has a positive role in increasing the accuracy of predicting potential reclaimed water availability as it takes full account of non-quantitative factors. However, the SEM-REM model is not perfect, because the model does not directly consider the public's acceptance of the development and utilization of reclaimed water, but only indirectly considers the acceptance of reclaimed water from the perspective of the level of reclaimed water treatment technology. Judging from the model prediction results, the lack of the public's acceptance of the development and utilization of reclaimed water has not had a catastrophic impact on the accuracy of the model's prediction. However, it is believed that full consideration of the uncertainty of the public's acceptance of the development and utilization of reclaimed water will further enhance the accuracy of the model. Since the model used in this paper has the advantage of requiring less statistical data, and overcomes the disadvantage of the traditional method of setting certain parameters with intense subjectivity, it has good general applicability. For other

countries, especially regions with relatively incomplete statistical data, a good prediction accuracy could be achieved by establishing a suitable SEM-REM model.

## 5. Conclusions

(1) For reclaimed water development and utilization, the driving factors outweigh the restraints, which indicates a high potential for development. Among the driving factors, the supply factor has the largest contribution (60%) to the development and utilization of reclaimed water, which is 10 times as large as that of the demand factor. Among the restraints, the lack of improved engineering conditions makes the largest contribution, followed by backward economic and technical conditions, and the water resource environment has the most negligible contribution.

(2) The structural equation model (SEM) was built to portray the inter-factor transmission pathways and quantify the contributions of influencing factors. Consequently, four indicators with the largest contribution to reclaimed water development and utilization ($\sum P \geq 68.8\%$) were identified, namely, the total amount of wastewater treated, the density of water pipelines in built-up areas, investment in facilities for reclaimed water treatment, and the processing of applications for water treatment patents. These four indicators could well explain the changes in the volume of reclaimed water utilization and hence could serve as the key indicators for predicting the potential availability of reclaimed water.

(3) With a good fitting accuracy ($R^2 = 0.847$), the established REM could effectively show how the empirical dot pitch fluctuated. Moreover, the potential prediction analysis found that China would maintain the rapid growth in reclaimed water utilization in the future, and up to 13.7 billion cubic meters of reclaimed water is expected to be available by 2025. This will help drive the efficiency of regional water recycling, while optimizing the structure of the urban water supply. It is worth noting that this paper used the SEM-REM model to calculate the amount of reclaimed water that could be developed and utilized in the future in China. By default, the reclaimed water quality meets the minimum water quality standards of the Chinese reclaimed water utilization regulations. In fact, the quality of the reclaimed water has different grades. The principle of high quality and superior use is very important for the graded development and utilization of reclaimed water. This involves a series of complex issues such as the choice of reclaimed water quality treatment process and the acceptability of water users for different levels of reclaimed water prices, which warrant further discussion in the future.

**Author Contributions:** Conceptualization and methodology J.Z., J.L. and T.M.; investigation: J.Z.; writing—draft preparation: J.Z.; writing—review and editing: J.Z., J.L., T.M., A.P. and X.D.; visualization: J.Z.; supervision: J.L. and T.M. All authors have read and agreed to the published version of the manuscript.

**Funding:** This work was supported by the second Tibetan Plateau Scientific Expedition and Research Program (Grant No. 2019QZKK0203) and the National Key R&D Program of China (Grant No. 2017YFC0403500).

**Institutional Review Board Statement:** Not applicable.

**Informed Consent Statement:** Not applicable.

**Data Availability Statement:** Not applicable.

**Acknowledgments:** We are thankful to anonymous reviewers and editors for their helpful comments and suggestions.

**Conflicts of Interest:** The authors declare no conflict of interest.

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
