# Peer review of "An SEM-REM-Based Study on the Driving and Restraining Mechanisms and Potential of Reclaimed Water Utilization in China"

_water, doi:10.3390/w14010052_

Round 1

Reviewer 1 Report

I have added my comments in the attached document.

Major concern - the lack of public perception consideration in the study.

The data presentation and depth is good, however, I wonder if Water journal is the best one to publish this paper as it contains mathematics that is different from current environmental-related journals. Perhaps a water management journal - just a suggestion.

Author Response

Response to Reviewer 1 Comments:

Point 1: Are these the authors' own developed equations? if not, references need to be added. If they are the authors', then a bit of explanation needs to accompany the equations.

Response 1: Agree and complete the modification. The relevant content involved in this amendment is formulas (1)~(3), and references had been added in the corresponding positions.

Point 2: Is this figure developed by the authors? the fonts appear as if this is a picture taken from somewhere.

Response 2: Agree and complete the modification. The relevant content involved in this amendment is Figure 1,which had been replaced with a new and clear picture in the corresponding position.

Point 3: The authors have missed a critical element for water reclamation, and that is public view. A number of literature has shown that, while technologically and economically water reuse is possible, lack of public acceptance has thwarted the move. 

Response 3: Agree and complete the modification. At the beginning of writing, the author had considered selecting some indicators to reflect the public’s acceptance of the development and utilization of reclaimed water. However, after consulting various data materials that had been officially issued, it was rare to find original statistics that could directly reflect the public's acceptance of the development and utilization of reclaimed water. Therefore, the author tried to find an index that could indirectly reflect the public's acceptance of the development and utilization of reclaimed water, that was, number of processed applications for water treatment patents. Generally speaking, the more advanced the technical level of the water treatment process, the higher the public acceptance. At the same time, judging from the model prediction results, the lack of the public’s acceptance of the development and utilization of reclaimed water has not had a catastrophic impact on the accuracy of the model’s prediction. However, it is believed that full consideration of the uncertainty of the public's acceptance of the development and utilization of reclaimed water will further enhance the accuracy of the model. Therefore, the author will further explore the impact of public acceptance on the development and utilization of reclaimed water in the future. The above content is mentioned in section 3.6 of the paper.

Point 4: How would these factors differ if the study was carried out outside China? All information is locally related and can thus have a small global following.

Response 4: Agree and complete the modification. Since the SEM-REM model used in this paper has the advantage of less statistical data needed, and overcomes the disadvantage of the traditional method of setting certain parameters with strong subjectivity, it has a good general applicability. For other countries, especially regions with relatively incomplete statistical data, a good prediction accuracy could be achieved by establishing a suitable SEM-REM model. The above content is mentioned in section 3.6 of the paper.

Point 5: One thing that catches the eye is that the majority of the references are China based. Not saying that these references are not valid, but the subject matter of this paper is of global concern and a number of studies has looked into the problem. Therefore I suggest that the authors suggest papers worldwide. Otherwise this paper will look more as a China-related paper.

Response 5: Agree and complete the modification. References had been added/adjusted according to the actual needs of the article.

Reviewer 2 Report

The aim of this study is very good. The manuscript has been well referenced. Study results may add to the existing knowledge. However, the following comments, besides English language improvement, may further enhance the quality of this paper:

  1. Three of the keywords duplicate the same as I the paper title. Other selections should be re-chosen.
  2. The last paragraph in the introduction section writes like the methodology of this study. It may be replaced with a specific objective statement of this study.
  3. All the methods used to analyze the problem matter have been described in detail. However, the procedures, steps and flow to fully analyze the objective of this study have not been clearly described (except just showing Figure 1) in the methodology section.
  4. A punctuation error (“,”) has been detected in line 2 under the sub-heading 3.2.1.
  5. Why not use “This paper” instead of “The paper”.
  6. The unit “100m m3” is very confusing. Why not use “100 Mm3”, and define “M” as “million”.
  7. Please include the currency equivalent of RMB and USD.
  8. Figure 4 is too detailed. It is not necessary to show all the “measured variables”, since they have already appeared in Table 1.
  9. Again Figure 5 is too small to read all the details. Why not create a new table to show all the correlation coefficients and skip all the “measured variables” in this figure.
  10. Figure 5 also seems to have two different figure captions.
  11. On the second line on Page 11, “Figure 4’ should be “Figure 5” instead.
  12. On Figure 6, the significant positive impact for E25 is highly questionable.
  13. There is no figure caption for Figure 8.
  14. Regression equations listed in the text do not match the equations shown in many figures.
  15. What is the unit used for “76.4 to 159.5” mentioned in the text? These values do not match the values shown in Figure 11.
  16. Discussion section is too brief. Why not merge with the Result section and rename the section as “Results and Discussions”.
  17. “Annual” is a better word for “year-on-year”.
  18. What is “Boasting high”? At what statistical significant level?
  19. This study only focused on the quantity. But, quality is also very important. This aspect should be mentioned in the conclusion remarks.

Author Response

Response to Reviewer 2 Comments:

Point 1: Three of the keywords duplicate the same as the paper title. Other selections should be re-chosen.

Response 1: Agree and complete the modification. In order to make it easier for readers to find the papers they need, keywords had been added/adjusted in this article.

Point 2: The last paragraph in the introduction section writes like the methodology of this study. It may be replaced with a specific objective statement of this study.

Response 2: Agree and complete the modification. The last paragraph had been appropriately modified to incorporate the specific objective statement, so that readers could more easily have a general understanding of this study.

Point 3: All the methods used to analyze the problem matter have been described in detail. However, the procedures, steps and flow to fully analyze the objective of this study have not been clearly described (except just showing Figure 1) in the methodology section.

Response 3: Agree and complete the modification. To make the explanation of the SEM model clearer, the author had explained the basic analysis process of the SEM model in more detail. The basic procedure of SEM analysis was subdivided into two stages: model development stage and model estimation and evaluation stage, and further explanations for each stage were expected to make it easier for readers to understand.

Point 4: A punctuation error (“,”) has been detected in line 2 under the sub-heading 3.2.1.

Response 4: Agree and complete the modification. The author had made the correct modification in the corresponding position.

Point 5: Why not use “This paper” instead of “The paper”.

Response 5: Agree and complete the modification. The author had made the correct modification in the corresponding position.

Point 6: The unit “100m m3” is very confusing. Why not use “100 Mm3”, and define “M” as “million”.

Response 6: Agree and complete the modification. The author had made correct changes in the corresponding positions of the text (including figures and tables).

Point 7: Please include the currency equivalent of RMB and USD.

Response 7: Agree and complete the modification. The author had made correct changes in the corresponding positions of the text (including figures and tables).

Point 8: Figure 4 is too detailed. It is not necessary to show all the “measured variables”, since they have already appeared in Table 1.

Response 8: Agree and complete the modification. For compact content, the author had deleted Figure 4.

Point 9: Again Figure 5 is too small to read all the details. Why not create a new table to show all the correlation coefficients and skip all the “measured variables” in this figure.

Response 9: Agree and complete the modification. To improve the reading quality of readers, Table 3 was added to this paper.

Point 10: Figure 5 also seems to have two different figure captions.

Response 10: Agree and complete the modification. The author had made the correct modification in the corresponding position.

Point 11: On the second line on Page 11, “Figure 4’ should be “Figure 5” instead.

Response 11: Agree and complete the modification. The author had made the correct modification in the corresponding position.

Point 12: On Figure 6, the significant positive impact for E25 is highly questionable.

Response 12: Agree and complete the modification. What the author wanted to express was that the coefficients of E25 and E26 were positive in each quantile, so there was a positive relationship with the development and utilization of reclaimed water. In order to avoid confusion when readers were reading, the relevant expressions had been adjusted.

Point 13: There is no figure caption for Figure 8.

Response 13: Agree and complete the modification. The author had made the correct modification in the corresponding position.

Point 14: Regression equations listed in the text do not match the equations shown in many figures.

Response 14: Agree and complete the modification. Because the author did not adjust the exact decimal place correctly when drawing, an error was generated. Now the relevant drawing had been modified.

Point 15: What is the unit used for “76.4 to 159.5” mentioned in the text? These values do not match the values shown in Figure 11.

Response 15: Agree and complete the modification. The unit of the relevant figure had been increased and the figure had been modified to the correct.

Point 16: Discussion section is too brief. Why not merge with the Result section and rename the section as “Results and Discussions”.

Response 16: Agree and complete the modification. The author appropriately increased the discussion part, and merged the discussion with the results in accordance with the revised requirements.

Point 17: “Annual” is a better word for “year-on-year”.

Response 17: Agree and complete the modification. The author had made the correct modification in the corresponding position.

Point 18: What is “Boasting high”? At what statistical significant level?

Response 18: Agree and complete the modification. The author considered that R2 of the fitting curve of the SEM-REM model reached to 0.847, which was an acceptable accuracy. However, in order to avoid misleading readers to think that 0.847 was a high accuracy, the author had modified some expressions in the corresponding position. In addition, the parameter estimation method in this paper used was point estimation, not interval estimation.

Point 19: This study only focused on the quantity. But, quality is also very important. This aspect should be mentioned in the conclusion remarks.

Response 19: Agree and complete the modification. The author fully agrees with the importance of water quality to the development and utilization of reclaimed water. To remind readers to be aware of this, the author had added some relevant content in the conclusion.

Round 2

Reviewer 1 Report

The authors have tried to address the concerns of the reviewer. As it stands now, the manuscript still misses a critical factor but that has been indicated in the new manuscript. 

Author Response

Point 1: The authors have tried to address the concerns of the reviewer. As it stands now, the manuscript still misses a critical factor but that has been indicated in the new manuscript.

Response 1: In order not to cause readers to ignore the key influencing factor of the public's acceptance of the development and utilization of reclaimed water, this paper emphasized the importance of this factor in section 3.6 of the paper. At the same time, the author will continue to actively explore how to quantitatively analyze the public's acceptance of the development and utilization of reclaimed water.

Reviewer 2 Report

An excellent job in revising the original manuscript. However, there is only one suggestion to further improve the quality of this paper and that is to carefully proof-read the entire paper to make sure the English and punctuation are absolutely correct. Point 4 of my previous comment has not been corrected.

Author Response

Response to Reviewer 2 Comments:

Point 1: An excellent job in revising the original manuscript. However, there is only one suggestion to further improve the quality of this paper and that is to carefully proof-read the entire paper to make sure the English and punctuation are absolutely correct. Point 4 of my previous comment has not been corrected.

Response 1: The spelling errors found by the expert had been corrected. In order to ensure that the paper no longer had low-level errors such as vocabulary spelling errors, grammatical errors, etc., all authors of this paper had been invited to read this paper carefully again. At the same time, grammatical errors and vocabulary spelling errors discovered during the reading process had been corrected.
